# Stromal Vascular Fraction Cell Therapy for a Stroke Patient—Cure without Side Effects

**DOI:** 10.3390/brainsci9030055

**Published:** 2019-03-06

**Authors:** Jaroslav Michalek, Alena Vrablikova, Karl Georg Heinrich, Zuzana Dudasova

**Affiliations:** 1Internal Consortium for Cell Therapy and Immunotherapy, Videnska 119, 619 00 Brno, Czech Republic; michalek@cellthera.org (J.M.); dudasova@cellthera.org (Z.D.); 2Cellthera Clinic, Videnska 119, 619 00 Brno, Czech Republic; 3Ordination DDr. Heinrich, Landhausgasse 2, 1010 Wien, Austria; office@ddrheinrich.com

**Keywords:** stromal vascular fraction, cell therapy, stroke

## Abstract

A 48-year-old male, who suffered from a stroke resulting in cerebellum damage and occlusion of the left vertebral artery, underwent stromal vascular fraction therapy. The clinical status of the patient was monitored by a modified Stroke Specific Quality of Life Scale before therapy and at 3, 9, 12, 18, 24, and 32 months after therapy. Three months after therapy, the patient felt a reduction in pain, vertigo, and fatigue. After 9 months, he was able to walk safely on his own. After 24 months, he was able to ride a bicycle. After 32 months, he felt completely healthy without any limitations or handicaps. Therefore, intravenous application of stromal vascular fraction cells represents a promising strategy for the treatment of patients after a stroke.

## 1. Introduction

Stroke is one of the most common cardiovascular diseases, affecting 4000–5000 out of 1 million people worldwide on an annual basis. More than one-third of patients die within a few days or weeks after having a stroke, while many others typically suffer from very serious consequences, including paralysis, speech problems, and other residual neurological deficits, for the rest of their lives [1]. Despite success in acute-phase treatment, including early thrombolysis, long-term care relies on physiotherapy and neurorehabilitation techniques, which generally offer a chance for improvement during the first 1–2 years of treatment, after which the patient typically reaches a plateau without further significant change in quality of life. Thus, novel treatment options for stroke victims are highly needed. Previously, we showed that adipose tissue-derived stromal vascular fraction (SVF) cells containing mesenchymal stromal cells have excellent regenerative potential [2,3]. Here, we demonstrate how autologous SVF therapy together with physiotherapy can cure a stroke victim without any side effects.

## 2. Methods

Our subject was a 48-year-old male stroke victim suffering from severe cerebellum damage and near-complete occlusion of the left vertebral artery. The patient was confined to a wheelchair due to balance loss, constant vertigo, fatigue, constant pain and frailty, smudged vision, and paralyzed esophagus with a failure to swallow. Even after 2 years of intense daily physiotherapy after the stroke, he was not capable of walking. Moreover, he suffered from persistent pain, vertigo, and frailty. After providing written informed consent, he underwent autologous SVF therapy. SVF cell isolation from adipose tissue was performed comparable to a method previously described [3]. Briefly, 200 mL of adipose tissue was obtained by standard liposuction, and the isolated SVF cells were resuspended in normal saline and applied intravenously during a single surgical procedure. The viability of the SVF cells was analyzed microscopically with nearly 95% viability. The clinical status of the patient was closely monitored by experienced physicians before SVF therapy and at 3, 9, 12, 18, 24, and 32 months after SVF therapy using a modified Stroke-Specific Quality of Life Scale (SS-QOL) [4].

## 3. Results

The case history of the 48-year-old male reveals that his quality of life, measured by the modified SS-QOL score, had reached a plateau 12 months prior to SVF therapy (Figure 1). During SVF therapy, the patient, weighing 94 kg and with a height of 175 cm (body mass index (BMI) of 30.69), was treated with approximately 180 million nucleated cells (i.e., 1.89 million SVF cells per kilogram of body weight) isolated from 200 mL of his adipose tissue. No postoperative pain, bleeding, or reactions to the local tumescent anesthesia were noted. After SVF therapy, he continued to engage in physiotherapy at a level of intensity similar to that prior to the procedure. It was recommended, however, that he lose weight by implementing a more plant-based diet. Just two weeks after SVF therapy, the patient reported feeling a reduction of pain and a mild reduction of vertigo. After 3 months, he experienced less fatigue. Moreover, slowly but steadily, he began to walk with support, demonstrating better balance. After 9 months, he was able to walk safely on his own. After 12 months, he was able to stand on one leg, and after 18 months, he was able to lose 18 kg of body weight, reaching a BMI of 28.08. After 24 months, he no longer suffered from balance problems, and he was able to start riding a bicycle again (for the first time in 4.5 years) with no fatigue. Finally, 32 months after SVF therapy, he reported feeling completely healthy with no limitations or handicaps (Figure 1).

## 4. Discussion

The application of autologous mesenchymal stromal/stem cells (MSCs), isolated from either bone marrow or adipose tissue, represents a new strategy for organ and tissue regeneration. MSCs can be easily and safely isolated from adipose tissue in much higher quantities, are more generically stable, and may be easily isolated by standard liposuction of adipose tissue, making them an ideal component of the stromal vascular fraction [2,5]. As we previously demonstrated, SVF cells have a huge tissue regeneration potential in patients with osteoarthritis [3,6]. Additionally, SVFs and adipose tissue-derived stem cells (ADSCs) have been clinically used in the treatment of multiple sclerosis [7], chronic myocardial ischemia [8], acute respiratory distress syndrome [9], Crohn’s disease [10], and other diseases. In the case of stroke victims, the standard treatment, including intense physiotherapy and neurorehabilitation, typically leads to improvements during the first few months or years after the stroke but then results in an eventual plateau regardless of the intensity of exercise [11]. Thus, SVF therapy represents a promising additional and complimentary treatment for stroke victims. Indeed, this strategy is gaining traction in the research community. In February 2019, Ouyang et al. published a meta-analysis of the safety and efficacy of stem cell therapies for ischemic stroke in preclinical and clinical studies [12]. Here, we contribute to this growing body of work by presenting a case study of successful SVF therapy for a stroke victim, which resulted in the complete curing of the patient with no remaining side effects.

## 5. Conclusions

Autologous SVF therapy of stroke is effective and safe method which significantly improves quality of life without side effects. This method represents promising strategy for patients with neurodegenerative diseases.

## Figures and Tables

**Figure 1 brainsci-09-00055-f001:**
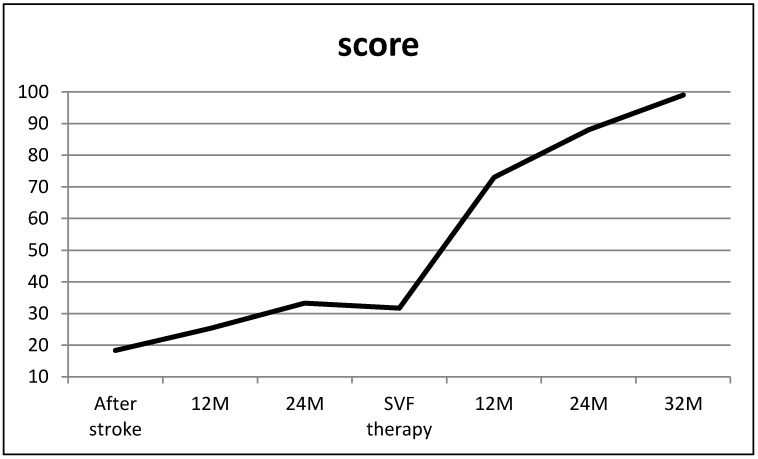
A total score analyzed by modified Stroke Specific Quality of Life Scale. The score was evaluated in the range from 1 to 100, where 100 reflect the patient’s complete health.

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
