# Peer review of "Stromal Vascular Fraction Cell Therapy for a Stroke Patient—Cure without Side Effects"

_brainsci, 2019, doi:10.3390/brainsci9030055_

Round 1
Reviewer 1 Report
The Authors described a nice case report on stromal vascular fraction cell therapy for a stroke patient. The topic is new, the results interesting. No evidence is provided of Ethical Committe approval; this is of the foremost importance, since the use of ADSCs for stroke is "off-label", and considered (in Europe) as cellular therapy. The following reference is missing: Raposio E, Caruana G, Bonomini S, Libondi G. A novel and effective strategy for the isolation of adipose-derived stem cells: minimally manipulated adipose-derived stem cells for more rapid and safe stem cell therapy. Plast Reconstr Surg. 2014 Jun;133(6):1406-9.
Author Response
Answer:
Informed Consent and Publishing Agreement signed by patient are attached. There is no Ethical Committee requirement in our country for one-day surgical procedures.
The references in discussion are accompanied.
English language and style were edited

Reviewer 2 Report
This study describes that a Stromal Vascular Fraction (SVF) therapy for a stroke patient with cerebellum damage and occlusion of the left vertebral artery, induces an entire cure. Indeed, there are questions that need to be answered or discussed: The authors isolated the SVF by liposuction. After washing and digestion, the SVF as a complex population contains diverse components (Gimble J and Katz A. circ. Research 2007) such as mesenchymal stem cells (MSCs), pericytes, circulating blood cells, endothelial cells, fibroblasts and adipose‐derived stromal cells (ASC). Did the authors deliver this complex population SVF or additionally isolated sub-fractions such as the ASC? If not, why the authors did not consider to deliver ASC known for tissue repair properties and commonly described as a therapy for a diverse human diseases? Did the authors determine the number of viable cells present in a cell suspension SVF? In this study, the patient was treated with his own SVF. Did the authors investigate the possibility to deliver the SVF from another patient? …To which extend there is risks of cellular rejection? How the authors can discuss this point? After the liposuction of the SVF, was there any postoperative pain, or treatment‐related adverse events, bleeding or reactions to anesthesia that have been noted? Considering the efficacy of cell therapy, can the authors comment on: whether the SVF could be more efficacious with a synergy effect of their components, OR isolated ASC / MSCs alone would be more efficacious? A typographical error in the line 10: … occlusion of the left vertebral…underwent. (The word “artery” is missing)
Author Response
English language and style were edited
Answer:
Complex population of SVF cells were delivered in intravenous infusion.
It has been shown that SVF may spontaneously form vessel-like networks in vitro and functional vasculatures in vivo. Such SVF could represent appropriate therapy for many diseases including stroke, ischemic disease or myocardial infarction. SVF cells are easily and quickly isolated from the body fat without the cultivation or other isolation associated with genetic instability or risk of contamination.The number of viable cells was analyzed microscopically with 94.7% viability.– 6. As only autologous SVF cells are being used, no risk of cellular rejection has been noticed in this patient and many others who were treated with autologous cells. We did not investigate the possibility of allogenic transplantation of SVF cells, but there is possibility of allogenic transplantation in people with low body fat level associated with risk of rejection of course. 7. None postoperative pain, bleeding or reactions to anesthesia were noted.8. We only perform infusions using freshly isolated SVF without culture expansion as there are potential risks including increased risk of infection in culture-expanded ASCs and potential risk of gene mutation when multiple passages of ASCs is performed. However, there is no evidence comparing the efficacy of autologous SVF and autologous ASCs for the treatment.
9. ”artery” accompanied in the text (line 10).